# Imaging Minimal Bacteria at the Nanoscale: a Reliable and Versatile Process to Perform Single-Molecule Localization Microscopy in Mycoplasmas

Fabien Rideau,[a] Audrey Villa,[a] Pauline Belzanne,[b] Emeline Verdier,[b] Eric Hosy,[b] ⬤ Yonathan Arfi[a]

[a]Université de Bordeaux, INRAE, Biologie du Fruit et Pathologie, UMR 1332, Villenave d'Ornon, France
[b]Interdisciplinary Institute for Neuroscience, CNRS, Université de Bordeaux, IINS, UMR 5297, Bordeaux, France

**ABSTRACT** Mycoplasmas are the smallest free-living organisms. These bacteria are important models for both fundamental and synthetic biology, owing to their highly reduced genomes. They are also relevant in the medical and veterinary fields, as they are pathogenic to both humans and most livestock species. Mycoplasma cells have minute sizes, often in the 300- to 800-nm range. As these dimensions are close to the diffraction limit of visible light, fluorescence imaging in mycoplasmas is often poorly informative. Recently developed superresolution imaging techniques can break this diffraction limit, improving the imaging resolution by an order of magnitude and offering a new nanoscale vision of the organization of these bacteria. These techniques have, however, not been applied to mycoplasmas before. Here, we describe an efficient and reliable protocol to perform single-molecule localization microscopy (SMLM) imaging in mycoplasmas. We provide a polyvalent transposon-based system to express the photoconvertible fluorescent protein mEos3.2, enabling photo-activated localization microscopy (PALM) in most *Mycoplasma* species. We also describe the application of direct stochastic optical reconstruction microscopy (dSTORM). We showcase the potential of these techniques by studying the subcellular localization of two proteins of interest. Our work highlights the benefits of state-of-the-art microscopy techniques for mycoplasmology and provides an incentive to further the development of SMLM strategies to study these organisms in the future.

**IMPORTANCE** Mycoplasmas are important models in biology, as well as highly problematic pathogens in the medical and veterinary fields. The very small sizes of these bacteria, well below a micron, limits the usefulness of traditional fluorescence imaging methods, as their resolution limit is similar to the dimensions of the cells. Here, to bypass this issue, we established a set of state-of-the-art superresolution microscopy techniques in a wide range of *Mycoplasma* species. We describe two strategies: PALM, based on the expression of a specific photoconvertible fluorescent protein, and dSTORM, based on fluorophore-coupled antibody labeling. With these methods, we successfully performed single-molecule imaging of proteins of interest at the surface of the cells and in the cytoplasm, at lateral resolutions well below 50 nm. Our work paves the way toward a better understanding of mycoplasma biology through imaging of subcellular structures at the nanometer scale.

**KEYWORDS** *Mycoplasma*, PALM, single-molecule localization microscopy, superresolution microscopy, dSTORM

**Ad Hoc Peer Reviewer** ⬤ Makoto Miyata, Osaka City University

Address correspondence to Yonathan Arfi, yonathan.arfi@u-bordeaux.fr.

The authors declare no conflict of interest.

The colloquial term "mycoplasmas" refers to a set of bacteria belonging to the *Mollicutes* class. These organisms derive from a common ancestor within the *Firmicutes* taxon through degenerative evolution that has led to an extreme reduction in genome size (~0.6 to 1.35 Mbp). During this process, mycoplasmas have lost a large number of

genes coding for important pathways, resulting in their characteristic lack of a cell wall and limited metabolic capacities (1–3). Owing to these deficiencies, mycoplasmas are obligate parasites that rely on their hosts for the production of a large array of essential metabolites. They have been isolated from a wide range of animals, including humans, mammals, reptiles, fish, and arthropods.

Mycoplasmas are the simplest self-replicating organisms known to date and are thought to be good representatives of a so-called "minimal" cell (4–6). They are therefore extremely interesting models in fundamental biology and have been used extensively to study the basic principles governing living systems and gene essentiality (7–11). These bacteria are also highly relevant in the field of synthetic biology, as their simplicity makes them prime models for the creation of engineered living systems. Mycoplasmas have been at the center of landmark studies, such as the production of the first cell governed by a chemically synthesized genome and, later, the first synthetic minimal bacterial cell (12, 13). Mycoplasmas are also the first cells for which complete and accurate predictive mathematical models have been developed (14–17).

In parallel to these fundamental aspects, mycoplasmas are also highly problematic organisms in both the medical and veterinary fields, as most of them are pathogenic for their hosts. In human, two species are particularly prevalent and concerning: *Mycoplasma genitalium*, which causes sexually transmissible urogenital infections (18), and *Mycoplasma pneumoniae*, which causes "atypical pneumonia," predominantly in children and immunocompromised patients (19). While both pathogens typically cause mild diseases with low mortality, these infections are often chronic and the pathogens are not completely eliminated after antibiotic treatments (20, 21). Mycoplasmas also infect most livestock species and are major pathogens of cows (*Mycoplasma bovis*) (22), goats and sheep (*Mycoplasma capricolum* subsp. *capripneumoniae*) (23), pigs (*Mycoplasma hyopneumoniae*) (24), and chickens (*Mycoplasma gallisepticum*) (25). Depending on the particular bacterial species or strain, mycoplasma infections can range from chronic, low-mortality inflammatory diseases to peracute, highly lethal diseases. Infected animals often display heavily reduced production yields, endangering farmers' revenues and threatening food security in poorly developed countries.

Studying mycoplasmas is a tedious process, as these organisms are slow growing and require complex and often undefined culture media. In addition, the number of genetic tools available is limited, except for a small set of species belonging to the *mycoides* cluster that has benefited from techniques derived from the aforementioned synthetic biology projects (26).

The physical size of mycoplasmas is also a key limiting factor, as most species have cells with dimensions in the 300- to 800-nm range. These values are close to the resolution of diffraction-limited optical microscopy, which is in the 200- to 300-nm range with commonly used dyes and high-numerical-aperture (NA) oil immersion objectives. Thus, fluorescence microscopy in mycoplasmas is often poorly informative, as it is extremely difficult to determine the subcellular localization of the imaged component. This problem exists for most bacteria and archaea and is exacerbated for mycoplasmas.

Higher-resolution techniques based on immunogold labeling and electron microscopy have therefore been preferred to localize proteins at the cell surface or in the cytoplasm of mycoplasma cells (27–31). However, these methods suffer from complex sample preparation protocols, are difficult to set up for simultaneous visualization of multiple molecular species, and are not compatible with live-cell imaging.

To date, only a few studies have used immunofluorescence to study protein localization in mycoplasmas, and all of them have focused on ascertaining the polar distribution of proteins in the cells of *Mycoplasma mobile*, *M. pneumoniae*, and *M. genitalium*, which all exhibit highly polarized shapes (32–35). Similarly, a small number of studies have reported the polar localization or colocalization of proteins fused to the fluorescent proteins mCherry and enhanced yellow fluorescent protein (EYFP), again in highly polarized cells (36–40). Other fluorescent proteins have been expressed successfully in several *Mycoplasma* species,

including green fluorescent protein (GFP) (41), Venus (42), mNeonGreen, and mKO2 (43), but have only been used as expression reporters or transformation markers.

Interestingly, the last decade has seen the rapid development of multiple new fluorescence microscopy techniques aimed at bypassing the diffraction limit and bridging the gap between optical imaging resolution and electron microscopy resolution. These methods, broadly termed single-molecule localization microscopy (SMLM), rely on the successive imaging of individual fluorophores to mathematically determine their exact position (44–46). The spatiotemporal decorrelation of fluorescence emissions can be achieved in a stochastic manner, through the use of specific dyes or fluorescent proteins that can be made to randomly emit their fluorescence under wide-field illumination. For instance, in the case of photo-activated localization microscopy (PALM), the protein of interest is fused to a specific fluorescent protein that can be either switched on or converted to another wavelength by UV illumination. Meanwhile, in the case of direct stochastic optical reconstruction microscopy (dSTORM), the protein of interest is labeled with a specific probe coupled to a fluorophore that has the ability to spontaneously transition into and out of a dark state under strong excitation illumination and in a reducing and oxygen-depleted buffer. SMLM techniques can typically yield images with lateral resolutions of $\sim$10 to 50 nm (47).

These methods have considerably expanded the possibilities of imaging in biology, allowing resolution of the subcellular organization of individual molecules or molecular assemblies, such as nuclear pores, chromatin complexes, and cytoskeletal filaments, at resolutions close to the molecular scale (48). The resolution improvements offered by SMLM appear especially attractive for microbiologists. These methods have been progressively adopted by the scientific communities, first through their establishment in model bacteria, such as *Escherichia coli* and *Bacillus subtilis*, and then by gradual transfer to more specialized fields (49–51). They have, however, not yet been applied to mycoplasmology.

In this report, we present the first protocols for the investigation of mycoplasmas using SMLM. We demonstrate the feasibility of PALM throughout the *Mollicutes* class by successfully expressing a photoconvertible fluorescent protein in six highly relevant *Mycoplasma* species. Then, working in the model organism *Mycoplasma mycoides* subsp. *capri*, we use PALM to image the subcellular localization of the cytoplasmic domain of an atypical F-type ATPase complex, yielding images with a lateral resolution of $\sim$40 nm. In parallel, we apply dSTORM in the same model to study the localization of a surface-anchored protease involved in virulence, yielding images at a lateral resolution of $\sim$25 nm.

## RESULTS

**Establishing a common and efficient sample preparation process for SMLM imaging in mycoplasmas.** The production of high-quality samples is generally regarded as a critical step for the acquisition of high-quality SMLM data, and multiple reviews provide important guidelines to follow (52, 53). Here, the first step of the process was to ensure the reliable immobilization of the mycoplasma cells on high-precision glass coverslips. We initially attempted to grow the mycoplasmas directly on poly-L-lysine-coated coverslips by immersing the coverslips in inoculated media. However, this approach failed, as the cells remained planktonic and did not adhere to the glass. We thus developed an efficient and reliable centrifugation-based process to force the cells to sediment and attach to the coverslip (Fig. S1 in the supplemental material). Briefly, mycoplasma cells are harvested and washed to form a homogenous medium-density suspension (approximately $10^5$ to $10^7$ CFU $\cdot$ mL$^{-1}$) in a 12-well plate in which the coverslip is immersed. Centrifugation at low speed in a swing-out rotor forces the cells to sediment and adhere to the glass. After washing to remove unbound cells, fixation is performed using a solution of 4% paraformaldehyde. The coverslip quality is checked by observation with a dark-field microscope. A good sample is characterized by bacterial cells deposited in a monolayer, regularly spaced and separated from each other by a few microns. These samples can either be imaged directly using PALM or further processed by immunolabeling in order to later perform dSTORM.

**Establishing a polyvalent mEos3.2 expression system for PALM in multiple mycoplasmas.** PALM imaging is based on the expression of specific fluorescent proteins that are either photoactivatable (irreversible off-to-on) or photoconvertible (wavelength A to wavelength B) (54). This process is driven by light and can be tuned to occur at a low rate, thus enabling imaging of individual fluorophores. Here, we elected to use the fluorescent protein mEos3.2 (55), which can be converted by illumination with a near-UV wavelength (405 nm) from a green state (excitation [Ex] = 507 nm and emission [Em] = 516 nm) to a red state (Ex = 572 nm and Em = 580 nm).

To assess the functionality of mEos3.2 in a wide range of *Mycoplasma* species, we first designed an mEos3.2 codon-optimized coding sequence using the codon usage table of *Mycoplasma mycoides* subsp. *mycoides* strain PG1 as the reference. This coding sequence was subsequently cloned in the plasmid pMT85 (43, 56–61). The plasmid backbone carries a transposon derived from Tn*4001* (62), the insertion of which can be selected through the gentamicin resistance gene *aacA-aphD* placed under the control of its natural promoter. In order to drive the expression of mEos3.2, we used the recently developed synthetic regulatory region PSynMyco (42). These three elements (transposon, selection marker, and SynMyco regulatory region) have all been shown to be functional in a wide range of *Mollicutes* species and should yield a universal *Mycoplasma* mEos3.2 expression vector.

The resulting plasmid, pMT85-PSynMyco-mEos3.2 (Fig. 1A), was used to transform six *Mycoplasma* species, relevant to either the veterinary or medical fields and covering the three main *Mollicutes* phylogenetic subgroups (Fig. 1B): (i) *M. mycoides* subsp. *capri* strain GM12 and *M. capricolum* subsp. *capricolum* strain 27343 from the *Spiroplasma* group; (ii) *M. bovis* strain PG45 and *Mycoplasma agalactiae* strain PG2 from the *hominis* group; and (iii) *M. genitalium* strain G37 and *Mycoplasma gallisepticum* strain S from the *pneumoniae* group. Transformants carrying the transposon were obtained for all six species, and sample coverslips were prepared for each.

We then assessed the ability of mEos3.2 to be converted to its red state by performing PALM imaging. Cells were observed in the red wavelength, with low-power illumination at 405 nm, to sparsely and stochastically photoconvert mEos3.2. Based on the acquired image stacks, the localization of each individual fluorescence emitter was determined through the PALMTracer plugin under MetaMorph, and then superresolved images were reconstructed and analyzed by automatic Voronoï-based segmentation of these localizations (Fig. 1C, Fig. S2). The principle of tessellation analysis is to extract objects with similar densities of localization. Here, with the first level of segmentation, we isolated the property of individual mycoplasma cells, with the second level yielding data on clustering of the protein of interest.

For all six *Mycoplasma* species studied, wild type and mutant, a comparable number of cells per field of view was observed in the transmission light (Fig. S3A). For each first-level segmented object, the number of underlying detections was compared between the wild-type and mEos3.2-expressing mycoplasmas. Similar results were obtained for each of the six *Mycoplasma* species studied (Fig. 1D). For the wild-type cells, only small numbers of individual objects were identified (*M. mycoides* subsp. *capri*, 240; *M. capricolum* subsp. *capricolum*, 291; *M. bovis*, 79; *M. agalactiae*, 99; *M. genitalium*, 110; and *M. gallisepticum*, 50), with each being supported by a small number of detections (median number of detections/object: *M. mycoides* subsp. *capri*, 42; *M. capricolum* subsp. *capricolum*, 31; *M. bovis*, 43; *M. agalactiae*, 63; *M. genitalium*, 61; and *M. gallisepticum*, 79). Among them, a large fraction of the objects were found outside the cells, suggesting that they were imaging artifacts rather than signals emitted by mEos3.2 (Fig. S3B). Comparatively, mEos3.2-expressing cells yielded 1 to 2 orders of magnitude more objects (*M. mycoides* subsp. *capri*, 1,487; *M. capricolum* subsp. *capricolum*, 1,183; *M. bovis*, 814; *M. agalactiae*, 3,761; *M. genitalium*, 919; and *M. gallisepticum*, 326), each supported by 1 order of magnitude more detections (median number of detections/object: *M. mycoides* subsp. *capri*, 245; *M. capricolum* subsp. *capricolum*, 132; *M. bovis*, 152; *M. agalactiae*, 237; *M. genitalium*, 493; and *M. gallisepticum*, 1,826)

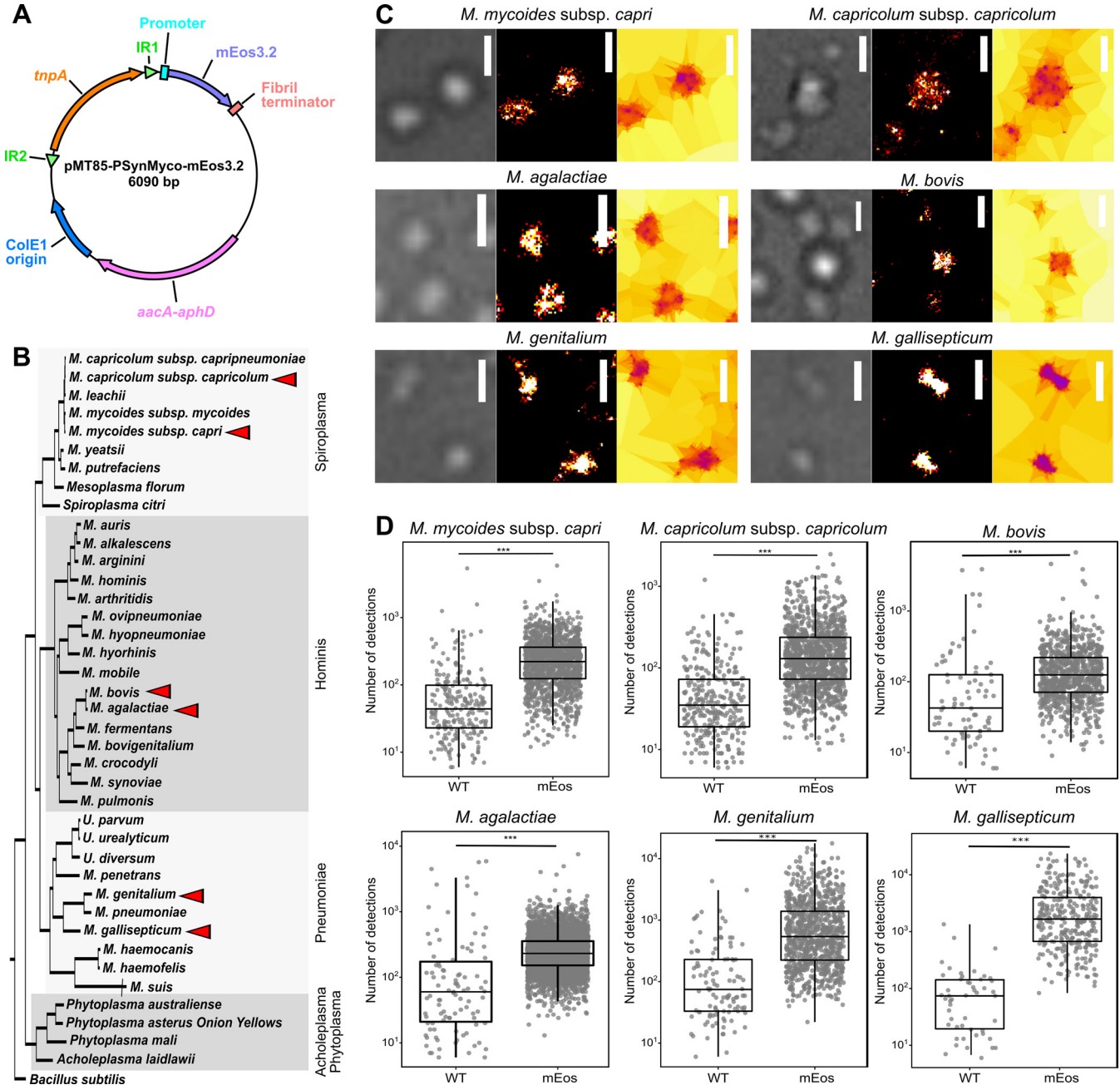

**FIG 1** Assessing the functionality of the photoconvertible fluorescent protein mEos3.2 in multiple *Mycoplasma* species. (A) Map of the plasmid pMT85-PSynMyco-mEos3.2. The main genetic components of the plasmid are indicated. IR, inverted repeat; *tnpA*, transposase; *aacA-aphD*, gentamicin resistance. (B) Distribution of the *Mycoplasma* species used in this study. A phylogenetic tree of representative *Mollicutes* species was inferred using the maximum-likelihood method from the concatenated multiple alignments of 79 proteins encoded by genes present at one copy in each genome (adapted from Grosjean et al. 2014 [99]). The main phylogenetic groups are indicated by gray boxes. The six species transformed with pMT85-PSynMyco-mEos3.2 are identified by red arrowheads. (C) Sample images of mycoplasma cells expressing mEos3.2 in their cytoplasm. For each of the six species, cells transformed with the plasmid pMT85-PSynMyco-mEos3.2 were imaged using PALM. A representative subset of the field of view is given (from left to right, phase-contrast image, superresolved reconstruction at 40-nm pixel size, and Tesseler segmentation of the localizations). Scale bars = 1 $\mu$m. (D) Quantification of the PALM signal intensity. For each of the six species, both wild-type (WT) and pMT85-PSynMyco-mEos3.2-transformed cells (mEos) were imaged by PALM, and the data collected from a single representative field of view (512 by 512 pixels; pixel size = 0.16 $\mu$m) were analyzed. The dot plot presents the number of detections measured in each object segmented by Tesseler (equivalent to a cell), on top of which a boxplot showing the median, interquartile range, minimum, and maximum values is overlaid. A statistical test (Mann-Whitney) was performed to compare the two conditions, WT and mEos. ***, $P < 1.10^{-10}$.

(Fig. 1D). In mEos-containing samples, most objects were found inside the boundaries of the cells (Fig. S3B). Taken together, the data collected indicate that mEos3.2 is functional in a wide range of *Mycoplasma* species, thus enabling PALM imaging of proteins of interest in these organisms.

**PALM imaging of an atypical F-type ATPase in *M. mycoides* subsp. *capri*.** To showcase the value of performing PALM to study biological processes in mycoplasmas, we worked in our model organism, *M. mycoides* subsp. *capri*. We were particularly interested in localizing an atypical F-type ATPase called $F_{1-like}$-$X_0$ (63), putatively involved in a mechanism of immune evasion (64). This ATPase is putatively formed by the assembly of two large domains: a membrane-anchored $X_0$ domain, formed by two transmembrane proteins and one cytoplasmic protein, and the $F_{1-like}$ domain, formed by four cytoplasmic proteins. The $F_{1-like}$ domain is predicted to be highly similar to the $F_1$ domain of the typical $F_1F_0$ ATP synthase and, thus, to comprise three alpha subunits and three beta subunits forming an alternating hexamer and a gamma and epsilon stalk connecting it to the $X_0$ domain. Leveraging the genome engineering tools available for this species, we generated a mutant strain, *M. mycoides* subsp. *capri* mEos3.2-0575, in which the fluorescent protein is fused to the N terminus of the $\beta$-subunit of the $F_1$-like domain. Diffraction-limited images taken by epifluorescence in the green wavelength (before photoconversion) did not yield any meaningful information, as they only showed dimly lit cells with no apparent distribution of the signal (Fig. 2A). In stark contrast, the superresolved images reconstructed from the PALM data sets revealed that most of the signal detected for each cell was localized in a small number of clusters that were found predominantly at the periphery of the cytoplasm (Fig. 2A). This localization was in accordance with preexisting information gathered on the association of the F domain of this ATPase with the internal face of the plasma membrane (63). The first level of tessellation successfully delineated the cells, and automatic segmentation of the localizations indicated that each cell typically presented 2 to 4 clusters of detections (mean of 3.3 and median of 3) (Fig. 2B). Meanwhile, cells expressing mEos3.2 as a monomer in the cytoplasm presented mainly a single cluster (Fig. S4). The median area of the clusters formed by mEos-tagged ATPase was 5,609 $nm^2$ (Fig. 2C), which corresponds to a circle with a diameter of 84 nm. This cluster area corresponds on average to ~2% of the median cell area, estimated to 324,518 $nm^2$ (equivalent to a circle 640 nm in diameter).

In order to estimate the accuracy of our detections, we used PALMTracer to track individual emitters and calculate the mean squared displacement (MSD) of the trajectories (Fig. 2D, inset). The fit of the first points of the MSD gives access to both the median speed of the molecule and its pointing accuracy (65). As our cells were fixed, we obtained the pointing accuracy of the fluorophore directly (66). The majority of pointing accuracies were between 30 and 60 nm, with a median of 41 nm (Fig. 2D). The imaging process was highly reproducible, with similar results obtained for data collected from two regions of the same coverslip or from two independent coverslips (median number of clusters per object, 3; median object area, 329,971 to 464,727 $nm^2$, equivalent to a circle with a diameter of 648 to 770 nm; median cluster area, 6,188 to 8,795 $nm^2$, equivalent to a circle with a diameter of 88 to 106 nm; median pointing accuracy, 39 to 51 nm) (Fig. S5).

**dSTORM imaging of a surface protease in *M. mycoides* subsp. *capri*.** In parallel to PALM, we also performed dSTORM imaging in *M. mycoides* subsp. *capri*. In this method, a fluorophore-conjugated probe, such as an antibody, is used to label the protein of interest for imaging. The fluorophores used in dSTORM have the ability to spontaneously transition into and out of a dark state under strong excitation illumination and in a reducing and oxygen-depleted buffer. To test this method in mycoplasmas, we first generated a mutant strain of *M. mycoides* subsp. *capri* in which the hemagglutinin (HA) epitope tag was fused to the C terminus of the protease $MIP_{0582}$ (64, 67). This immunoglobulin-specific protease is anchored to the cell surface and belongs to the same immune evasion system as the F-type ATPase studied above. Samples of *M. mycoides* subsp. *capri* 0582-HA on coverslips were immunolabeled with a mouse anti-HA primary antibody and a goat anti-mouse Alexa Fluor 647-conjugated secondary antibody. Acquisition of diffraction-limited data was performed by illuminating the sample with the excitation laser at a low power that was not sufficient to cause blinking of the fluorophores. Again, no meaningful information

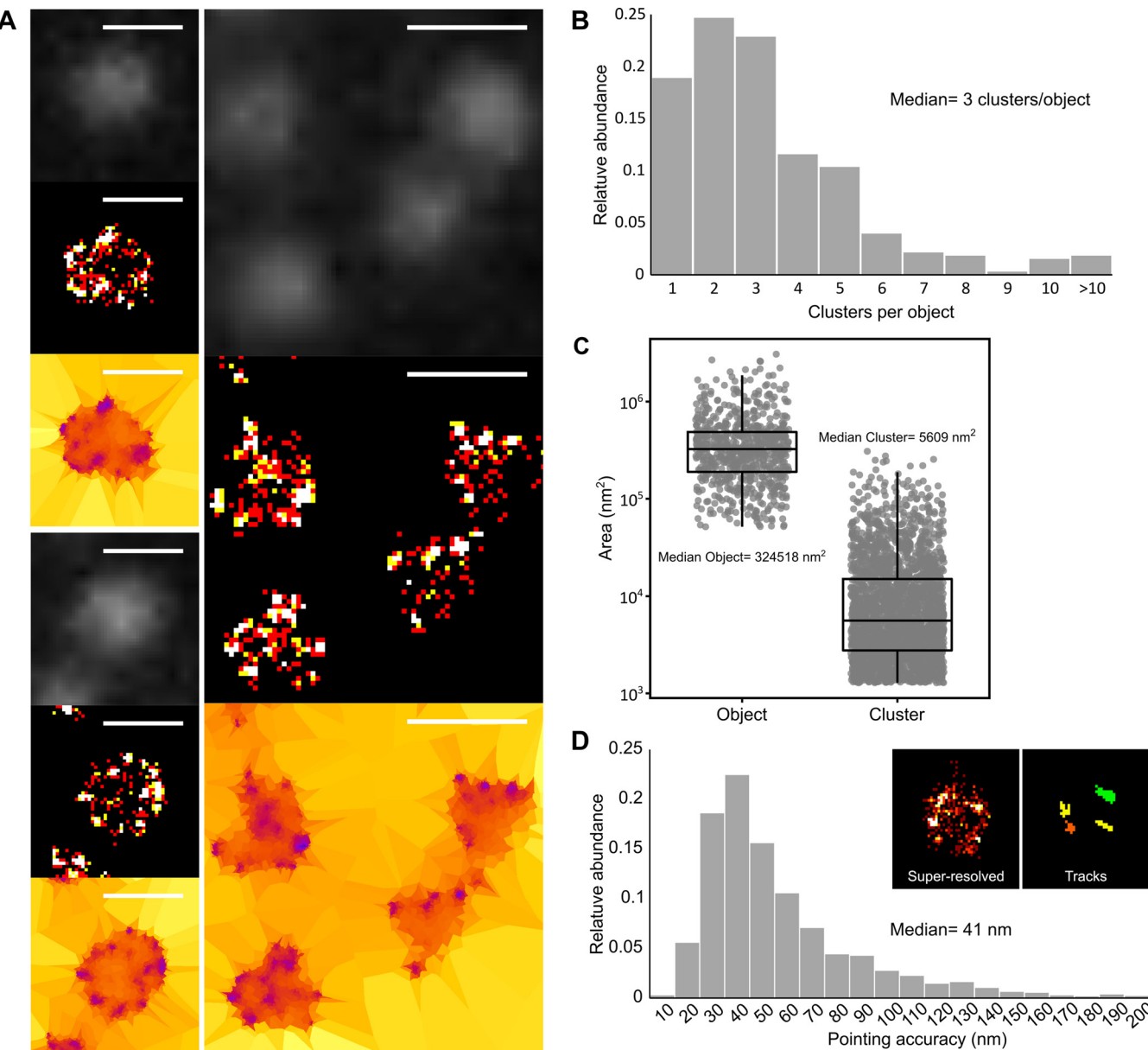

**FIG 2** PALM imaging of an F-type ATPase in *Mycoplasma mycoides* subsp. *capri*. *M. mycoides* subsp. *capri* mEos3.2-0575 cells, expressing a mEos-fused variant of the $\beta$-subunit of the ATPase $F_{1-like}$ domain, were imaged by PALM. In this *M. mycoides* subsp. *capri* mutant, the fluorescent fusion protein is expressed from the native genomic locus and replaces the wild-type variant. The data presented here correspond to a single representative field of view (512 by 512 pixels; pixel size = 160 nm). (A) Sample images of *M. mycoides* subsp. *capri* mEos3.2-0575 cells. For each field of view, the images correspond to epifluorescence (diffraction limited) (top), superresolved reconstruction (40-nm pixel) (middle), and Tesseler segmentation (bottom). Scale bar = 1 $\mu$m. (B) Tesseler clustering of the fluorescence signal. The number of clusters per Tesseler-segmented object was computed. The bar graphs display the distribution of the numbers of clusters per object. (C) Object and cluster sizes. The dot plot presents the area (in $nm^2$) of each object and cluster segmented by Tesseler, to which a boxplot showing the median, interquartile range, minimum, and maximum values is overlaid. The median value of each data set is indicated. (D) Evaluation of the PALM imaging pointing accuracy. Inset, example of the tracks computed using PALMTracer from which the $MSD_0$ and pointing accuracy values are derived. The bar graphs display the distribution of the pointing accuracies derived from each track. The median value of the data set is indicated.

could be extracted, as the cells appeared dimly fluorescent, with a slightly more intense signal at the periphery in some cases (Fig. 3A). Conversely, superresolved images reconstructed from the dSTORM data sets revealed that the fluorescence signal was localized in clusters that were predominantly at the periphery of the cell (Fig. 3A). This was in accordance with the previously demonstrated localization of the protease at the cell surface (64).

Automatic segmentation of these localizations indicated that cells typically presented 5 to 10 clusters (median of 6) and a significant proportion of cells exhibited

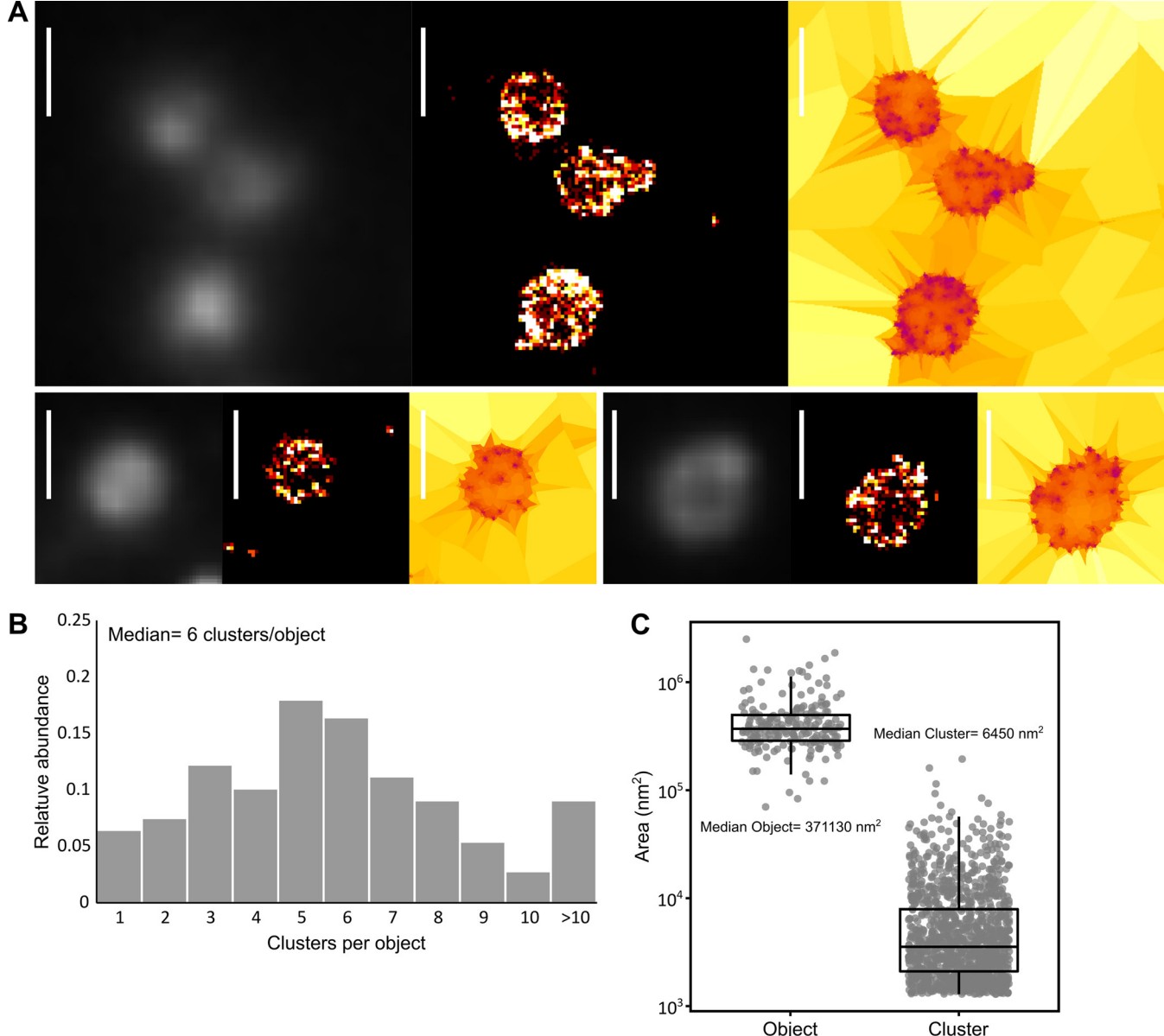

**FIG 3** dSTORM imaging of an antibody-specific protease in *Mycoplasma mycoides* subsp. *capri*. *M. mycoides* subsp. *capri* 0582-HA cells expressing an HA tag-fused variant of the serine protease MIP$_{82}$ were immunolabeled and imaged by dSTORM. The tagged fusion protein is expressed from the native genomic locus and replaces the wild-type variant. The data presented here correspond to a single representative field of view (512 by 512 pixels; pixel size = 160 nm). (A) Sample images of *M. mycoides* subsp. *capri* 0582-HA cells. For each field of view, the images correspond to epifluorescence (diffraction limited) (left), superresolved reconstruction (40 nm pixel) (middle), and Tesseler segmentation (right). Scale bar = 1 $\mu$m. (B) Tesseler clustering of the fluorescence signal. For each field of view, the number of clusters per Tesseler-segmented object was computed. The bar graphs display the distribution of the numbers of clusters per object. (C) Object and cluster sizes. The dot plot presents the area (in nm$^2$) of each object and cluster segmented by Tesseler, to which a boxplot showing the median, interquartile range, minimum, and maximum values is overlaid. The median value of each data set is indicated.

more than 10 clusters (10%) (Fig. 3B). The median cluster area was 3,254 nm$^2$ (Fig. 3C), corresponding to a circle with a diameter of 64 nm. Interestingly, these clusters were smaller than those measured by PALM for the ATPase, with a median area inferior by ~40% and a corresponding circle diameter inferior by ~25%, probably due to the higher pointing accuracy obtained with the dSTORM technique (25 nm) than with PALM (41 nm). However, the first level of tessellation objects, which corresponds to the cells, were very similar to those observed by PALM, with a median area of 371,130 nm$^2$ (equivalent to a circle with a diameter of 687 nm). Again, the imaging process showed high reproducibility, with similar results from data collected from two regions of the same coverslip or from two independent coverslips (median number of clusters per

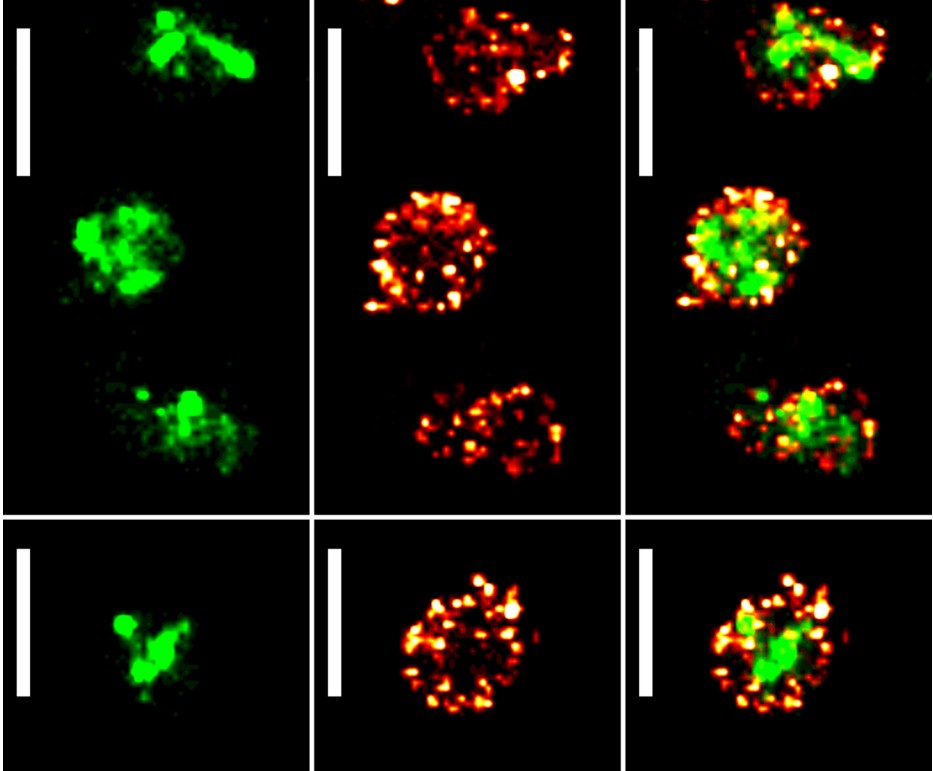

**FIG 4** PALM/dSTORM two-color imaging of *Mycoplasma mycoides* subsp. *capri*. Sample images of *M. mycoides* subsp. *capri* 0582-HA pMT85-PSynMyco-mEos3.2 cells, expressing both an HA tag-fused variant of the serine protease MIP$_{0582}$ and the fluorescent protein mEos3.2. The tagged fusion protein is expressed from the native genomic locus and replaces the wild-type variant. mEos3.2 is expressed from a transposon inserted at a random site in the bacterial chromosome. For each field of view, the images correspond to a reconstructed PALM image (40-nm pixel) (left), a reconstructed dSTORM image (40-nm pixel) (middle), and an overlay of the reconstructed PALM and dSTORM images (right). Scale bar = 1 $\mu$m. All the images were sampled from the same coverslip and field of view.

object, 5 to 7; median object area, 353,506 to 422,145 nm$^2$, equivalent to a circle with a diameter of 670 to 733 nm; median cluster area, 3,582 to 3,944 nm$^2$, equivalent to a circle with a diameter of 67 to 70 nm) (Fig. S6).

**Dual-color and three-dimensional SMLM in *M. mycoides* subsp. *capri*.** One of the key advantages of fluorescence imaging is the ability to perform multicolor experiments to locate multiple proteins of interest by combining fluorophores with different excitation/emission spectra. Here, we leveraged the spectral compatibility between the mEos3.2 red state and the Alexa Fluor 647 emission/excitation wavelengths to perform dSTORM/PALM dual-color imaging in *M. mycoides* subsp. *capri*. To do so, the strain *M. mycoides* subsp. *capri* 0582-HA was transformed with the plasmid pMT85-PSynMyco-mEos3.2. The transformants were then imaged sequentially, first by dSTORM and then by PALM. Data sets were processed as described above. Superresolved images were successfully reconstructed from both data sets, yielding results similar to those observed during single-color imaging: mEos filling the cytoplasm in PALM and 10 to 20 small clusters of MIP$_{0582}$-HA at the cell periphery in dSTORM (Fig. 4A). This experiment demonstrates the ability of superresolution techniques to identify different types of organization in the minute mycoplasma cells, while classical microscopy failed.

**Using mEos3.2 as a reporter and PALM imaging to measure physical parameters.** In addition to the acquisition of localization data for proteins of interest, we also demonstrated that the PALM imaging process could be used to gather information on protein expression levels. Indeed, similar to what can be done with classical fluorescent proteins, the amount of mEos3.2 in a given cell is proportional to the number of detections. We compared the relative strengths of various promoters by building modified versions of the plasmid pMT85-PSynMyco-mEos3.2 in which the PSynMyco promoter

was replaced by the P438 or PSpi promoter. The three plasmids were transformed into *M. mycoides* subsp. *capri*, yielding transformants that were subsequently imaged by PALM. Analysis of the data revealed that each promoter drove the expression of mEos3.2 at a different level (Fig. 5A), with P438 producing a signal slightly above the background noise (P438, 46 detections/object, versus wild type, 23 detections/object), PSynMyco producing a signal 1 order of magnitude higher (104 detections/object), and PSpi giving the highest signal (858 detections/object).

Interestingly, the overexpression of mEos3.2 alone can also be used to estimate the absolute physical size of individual cells. Indeed, for each of the hundreds to thousands of cells found in the field of view, the cytoplasm could be reliably discriminated from the background based on the localization of the fluorescence signal. The surface occupied by the cytoplasm could then be accurately measured (length, width, and area) (Fig. S7). As an example, we analyzed a population of 358 cells of *M. mycoides* subsp. *capri* pMT85-PSpi-mEos3.2 (Fig. 5B), showing a normal distribution of sizes ranging from 107 to 1,490 nm (mean of 768 nm and median of 754 nm) on the major axis and 87 to 1,179 nm (mean of 540 nm and median of 540 nm) on the minor axis.

## DISCUSSION

Superresolution microscopy techniques have revolutionized biology by enabling fluorescence imaging well beyond the diffraction limit. Establishing these methods in mycoplasmas represents a significant milestone for the investigation of these minute bacteria with atypical properties and broad biological relevance.

Here, we report the first application of PALM and dSTORM, two mature SMLM techniques, in mycoplasmas. We describe a common, simple, and reliable method to generate high-quality, SMLM-ready, fixed samples and showcase the value of superresolution imaging in our model species *Mycoplasma mycoides* subsp. *capri*. This bacterium, in addition to being the etiologic agent of pulmonary infections in goats, has emerged as a major model in both synthetic and fundamental biology. Indeed, it is the original organism from which the synthetic cells JCVI Syn1.0 and Syn3.0 have been derived through genome engineering and reduction (12, 13). As a result, the SMLM strategies developed here in *M. mycoides* subsp. *capri* are likely to be applicable in its man-made descendants.

We imaged two proteins of interest in *M. mycoides* subsp. *capri*, one cytoplasmic and one at the cell surface, with a pointing accuracy below 50 nm (more than 6 times the diffraction limit). The images reconstructed from our PALM and dSTORM data sets provided qualitative and quantitative data that were previously only accessible through immunogold electron microscopy. Our imaging process proved fast, as sample preparation took ~3 h for PALM and ~6 h for dSTORM, followed by ~10 to 15 min of data acquisition for each field of view. Analysis of the data generated during our PALM and dSTORM experiments was performed using a selection of software in use in our laboratories at the time. It should be noted that a wide array of alternative data analysis approaches is available for each step of the process (localization algorithms, superresolution image rendering, clustering, colocalization, etc.). Available options are well documented in the literature (68–71).

Given that each field of view contains several hundred to several thousand cells, it is possible to collect data at a medium throughput and to get an accurate description of a biological process across a large population of cells. In addition, we note that our process is highly reproducible, with similar results obtained across multiple fields of view acquired on the same coverslip or across independent coverslips, prepared from independent biological replicates and imaged on different dates.

In this study, we focus our analysis on the localization of virulence factors associated with the cell membrane. However, SMLM can be applied to study any protein of interest, including those involved in fundamental biological processes like transcription, translation, or cell division, and has the potential to become a key tool for the mycoplasmologist community. In order to foster a quick adoption of SMLM, we have validated our sample preparation process in a total of six different *Mycoplasma* species,

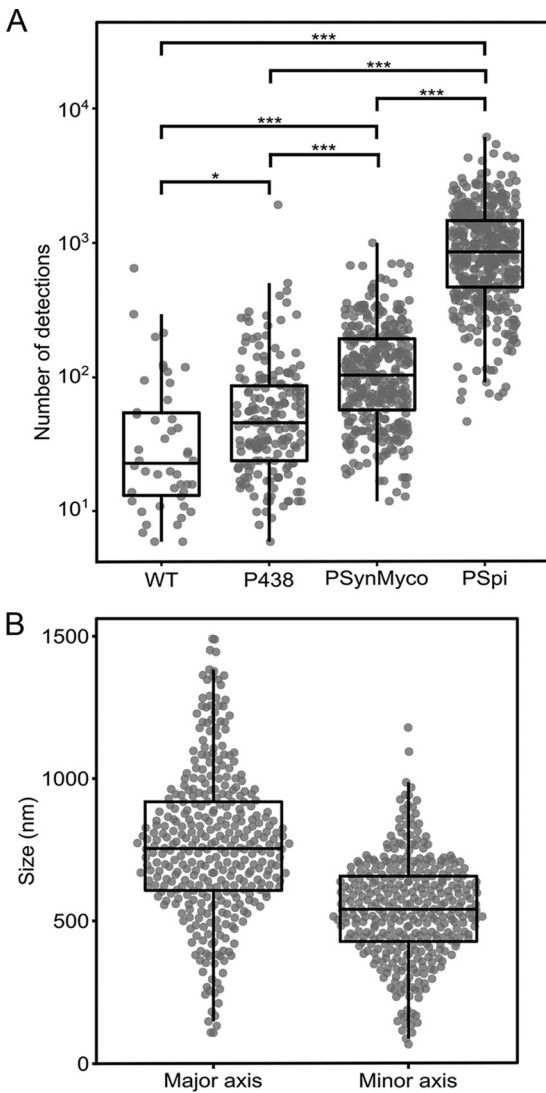

**FIG 5** Evaluation of promoter strength and cell size through PALM imaging in *M. mycoides* subsp. *capri*. (A) Comparison of promoter strength by PALM imaging. *M. mycoides* subsp. *capri* cells, either wild type (WT) or transformed with plasmid pMT85-P438-mEos3.2 (promoter P438), pMT85-PSynMyco-mEos3.2 (promoter PSynMyco), or pMT85-PSpi-mEos3.2 (promoter PSpi), were imaged by PALM. For each strain, the data collected from a single representative field of view (512 by 512 pixels; pixel size = 0.16 $\mu$m) were analyzed. The dot plot presents the number of detections measured in each object segmented by Tesseler (equivalent to a cell), on top of which a boxplot showing the median, interquartile range, minimum, and maximum values is overlaid. Statistics tests (Mann-Whitney) were performed to compare the four strains (*, $P < 0.05$; ***, $P < 1.10^{-10}$). (B) Deriving cell size data from PALM images. *M. mycoides* subsp. *capri* cells transformed with plasmid pMT85-PSpi-mEos3.2 were imaged by PALM. The data collected from a single representative field of view (512 by 512 pixels; pixel size = 0.16 $\mu$m) were analyzed. The dot plot presents the dimensions (in nm) of the major axis and the minor axis of each object segmented by Tesseler (Fig. S7), on top of which a boxplot showing the median, interquartile range, minimum, and maximum values is overlaid.

relevant to both the medical and veterinary fields and covering the three main *Mollicutes* phylogenetic subgroups.

We also provide the plasmid pMT85-PSynMyco-mEos3.2, which allows the expression of the photoconvertible fluorescent protein mEos3.2 under the control of the SynMyco regulatory element, enabling PALM imaging in all of the *Mycoplasma* species tested. We also produced two variants of the plasmid in which the mEos3.2 expression is driven by the alternative promoters PSpi (72) and P438 (73), which are well known and used throughout the *Mollicutes* field. These three plasmids are available through the repository Addgene (catalog numbers 173894, 173895, and 173896) for rapid distribution and for evaluation in other species. It should be noted that mEos3.2 is generally

regarded as one of the best fluorophores for PALM imaging, as it exhibits multiple traits that are desirable for SMLM: it is monomeric and has fast maturation and relatively high photostability. Its green and red excitation and emission wavelengths also make it compatible with widely used DNA dyes like Hoechst 33342 (361/497 nm), membrane dyes like Nile red (552/636 nm), and far-red organic dyes like Alexa Fluor 647 (650/670 nm) for multicolor imaging. Alternative fluorescent proteins could be expressed from our pMT-85 backbone, and multiple options are available in the literature, including the well-characterized mMaple3 (74), PAmCherry (75), or Dendra2 (76). These alternatives should nonetheless be properly benchmarked, as it was shown that the performances of different fluorescent proteins can vary greatly in a given microbial species (77). In addition, fusion to a fluorescent protein carries the inherent risk of causing negative effects on the protein of interest (improper folding, oligomerization, or reduced stability) or on the cell (cytotoxicity or altered behavior) (54).

It is noteworthy that our PALM experiments were performed on fixed samples and, thus, did not leverage an important benefit of this technique: the ability to image live cells (78). This choice was constrained by biosafety requirements, as most mycoplasmas are biosafety level 2 (BSL2) organisms. Live-cell imaging would have required the microscopy set-up to be located in an appropriate laboratory, which was not available to us. This issue will probably affect most mycoplasmologists, as BSL2-confined SMLM setups are currently rare worldwide. In addition, SMLM imaging equipment is predominantly housed in laboratories working with eukaryotic cell cultures. As mycoplasmas are frequent contaminants of these cultures (79), mycoplasmologists might be barred from accessing these facilities with live cells. Transition to live-cell PALM is nonetheless desirable, as it would enable imaging of dynamic processes, offsetting the relative lack of resolution of the technique by providing more physiologically relevant data. Live-cell imaging will also require an alternative sample preparation process, as adhering to the poly-L-lysine-coated coverslip might have a negative impact on the cell growth of normally planktonic species. Protocols revolving around sandwiching bacterial cells between agarose pads are well documented (77, 80) and could be rapidly adapted to mycoplasmas.

Meanwhile, in order to perform dSTORM, we relied on the expression of an epitope-tagged variant of our protein of interest that was subsequently immunolabeled. This choice was guided by the possibility of using commercial, well-characterized, high-affinity monoclonal anti-HA antibodies, enabling us to have good labeling of the protein of interest and low background noise. However, genetic edition strategies allowing the precise modification of a genomic locus for epitope tagging are only available in a limited number of *Mycoplasma* species. This issue can, however, be bypassed by using custom, protein-specific antibodies. It is noteworthy that our protocol used a primary antibody and a fluorescent-protein-labeled secondary antibody. This method, while practical and widely used, introduces a significant displacement between the targeted protein and the reporting fluorophore due to the physical size of the antibodies (81). This linkage error is approximately 15 to 20 nm and can be mitigated through the utilization of either a fluorophore-conjugated primary antibody or alternative small probes, such as nanobodies or aptamers, which offer an ~5-nm linkage error (82–84). Finally, it should be noted that superresolution microscopy is still a developing field, which sees continuous improvements in resolution through new techniques. For instance, strategies based around expansion microscopy (a process in which the sample is physically expanded in an isotropic fashion) could help reach the nanometer scale by decrowding biomolecules in dense samples (85). Meanwhile, the recently developed MINFLUX microscope promises imaging at 1- to 3-nm lateral resolution, in multicolor and in three dimensions (86). These improvements, coupled with the commercialization of user-friendly microscopy systems, will further drive the adoption of SMLM and in the long term will benefit our understanding of mycoplasma biology.

## MATERIALS AND METHODS

**Bacterial strains and culture conditions.** *Escherichia coli* strain NEB-5$\alpha$, used for plasmid cloning and propagation, was grown at 37°C in LB medium supplemented with antibiotics (10 $\mu$g/mL tetracycline, 50 $\mu$g/mL kanamycin, and 100 $\mu$g/mL ampicillin) when selection was needed.

*Mycoplasma mycoides* subsp. *capri* strain GM12, *Mycoplasma capricolum* subsp. *capricolum* strain 27343, *Mycoplasma bovis* strain PG45, *Mycoplasma agalactiae* strain PG2, *Mycoplasma genitalium* strain G37, and *Mycoplasma gallisepticum* strain S6 were grown at 37°C in the appropriate medium, either SP5 (72), Hayflick modified (87), or SP4 modified (87, 88), supplemented with antibiotics (100 to 400 $\mu$g/mL gentamicin and 5 $\mu$g/mL tetracycline) when selection was needed.

**Cloning the mEOS3.2 expression plasmids.** The mycoplasma codon-optimized version of the coding sequence for mEos3.2 was designed using the online tool JCat (http://www.jcat.de/) (parameters: input = protein sequence, reference organism = *Mycoplasma mycoides* [subsp. mycoides SC, strain PG1], options = "Avoid prokaryotic ribosome binding sites"). Further adaptation of the codon usage was performed manually based on the recommendations of the Twist Bioscience synthesis tool, to enable the chemical synthesis of the corresponding DNA fragment (Twist Bioscience). After PCR amplification, the coding sequence was cloned by InFusion (Clontech) in the plasmid pMT85 under the control of the promoter PTufA (43). Based on this pMT85-PTufA-mEos3.2 plasmid, three other variants were subsequently produced by Gibson assembly (NEB) or by site-directed mutagenesis (NEB) to replace the original promoter with the promoter PSpi (72), the promoter PSynMyco (42), or the promoter P438 (73). Cloned plasmids were transformed into chemically competent *E. coli* NEB-5$\alpha$ cells for maintenance and propagation and then isolated by miniprep, verified by enzymatic restriction, and finally checked by Sanger sequencing (Genewiz).

**Mycoplasma transformation.** Plasmid pMT85-PSynMyco-mEos3.2, pMT85-PSpi-mEos3.2, or pMT85-P438-mEos3.2 was transformed by polyethylene glycol contact into *M. mycoides* subsp. *capri*, and *M. capricolum* subsp. *capricolum* (89), *M. agalactiae* and *M. bovis* (90), and *M. gallisepticum* (unpublished data) or by electroporation into *M. genitalium* (56). Transformants were subsequently plated on appropriate medium containing antibiotics for selection (see above). Transformant clones were then passaged 3 times in liquid medium supplemented with gentamicin (100 to 400 $\mu$g/mL). The presence of the mEOS3.2 coding sequence was checked by PCR.

**Production of *Mycoplasma mycoides* subsp. mycoides mutants.** Accurate editing of the genome of *M. mycoides* subsp. *capri* is currently only possible through a complex process involving the transfer of the bacterial chromosome into a yeast cell, its modification in the yeast, and its subsequent transplantation back into an *M. capricolum* subsp. *capricolum* bacterial recipient cell (64). This process was used to generate two mutant strains: *M. mycoides* subsp. *capri* 0582-HA, in which an HA tag (91) coding sequence is fused to the C terminus of the coding sequence of MMCAP2_0582, and *M. mycoides* subsp. *capri* mEos3.2-0575, in which the codon-optimized mEos3.2 coding sequence is fused to the N terminus of the coding sequence of MMCAP2_0575 (see the supplemental material). The HA tag was placed at the C terminus of MMCAP2_0582 due to the presence of a membrane anchoring domain in the N terminus and because previous studies showed that MIP tolerates the presence of small tags in the C terminus (64, 67). Meanwhile, mEos3.2 was fused to the N terminus of the beta subunit of the ATPase, as previous studies have shown that this location is amendable to fusion and the addition of large molecular structures (92, 93).

To produce the mutants, we first generated two plasmids encoding guide RNAs targeting either MMCAP2_0582 or MMCAP2_0575 by modification of the base plasmid p426-SNR52p-gRNA.Y-SUP4t (26) using the Q5 site-directed mutagenesis kit (NEB). In addition, we generated two recombination cassettes that contained the modified version of the target loci flanked by 1,000-bp recombination arms. Both cassettes were built by first cloning the wild-type locus and 1-kb flanking region in the pGEM-T plasmid (Promega) and subsequently modifying the plasmid to add the HA tag coding sequence using the Q5 site-directed mutagenesis kit (NEB) or the mEos3.2 coding sequence using InFusion (Clontech). Plasmids carrying the cassettes were checked by Sanger sequencing (Genewiz), and the cassettes amplified by PCR. Both the guide RNA-encoding plasmid and the recombination cassette were then cotransformed into the yeast carrying the *M. mycoides* subsp. *capri* chromosome. Yeast transformants were checked for the presence of the integral bacterial chromosome by multiplex PCR and for the presence of the desired mutation by PCR and amplicon sequencing. The modified genome was then extracted and transplanted into the recipient cell. The resulting transplants were also checked for genome integrity by multiplex PCR and for the presence of the desired mutation by PCR and amplicon sequencing.

**Sample preparation for SMLM experiments.** Mycoplasma cells, grown to late log phase at 37°C in an appropriate medium supplemented with gentamicin (100 to 400 $\mu$g/mL), were harvested by centrifugation at 6,800 relative centrifugal force (rcf) at 10°C for 10 min. After removal of the spent medium, the cells were washed twice in one volume of buffer (67.7 mM HEPES, 140 mM NaCl, 7 mM MgCl$_2$, pH 7.35) and subsequently resuspended in 1/3 volume of the same buffer. When necessary, cell aggregates were broken down by performing 20 passages through a 26-gauge needle. A poly-L-lysine-coated no. 1.5H 18-mm precision coverslip (Marienfeld) was placed at the bottom of a 12-well plate and equilibrated in 1 mL of wash buffer. Three microliters of the cell suspension was added to the well, and the plate was subsequently centrifuged in a swing-out rotor at 2,500 rcf at 10°C for 10 min to force the cells to sediment on the coverslip. The well was emptied by suction, and the coverslip was then moved to a clean well. The coverslip was then washed once with 3 mL wash buffer and then incubated in 1 mL of prewarmed 4% paraformaldehyde (PFA) in washing buffer at 37°C for 30 min. Five washing steps with 3 mL of washing buffer were performed to eliminate all traces of PFA. Correct deposition and fixation of the cells on the coverslip was checked by dark-field microscopy (Nikon Eclipse Ni microscope equipped with a dark field condenser and a Teledyne Photometrics Iris 9 camera) using washing buffer as the mounting medium. Validated coverslips were stored at 4°C in phosphate-buffered saline (PBS; 137 mM NaCl, 2.7 mM KCl, 10 mM Na$_2$HPO$_4$, 1.8 mM KH$_2$PO$_4$) until ready to use for PALM imaging (no more than 10 days).

For dSTORM imaging, coverslips were prepared as described above. After the final wash, each coverslip was incubated for 1 h at room temperature in 1 mL of blocking buffer (PBS/1% bovine serum

albumin [BSA]). Immunolabeling was performed by first using a mouse anti-HA tag primary antibody (Thermo) diluted at 1/500 in blocking buffer. After incubation for 1 h at room temperature, the coverslip was washed three times with 1 mL of PBS. Then, the coverslip was incubated for 1 h at room temperature with the secondary antibody (goat anti-mouse IgG conjugated to Alexa Fluor 647; Jackson ImmunoResearch) diluted at 1/500 in 1 mL of blocking buffer. After three wash steps in PBS, an additional fixation was performed by incubating the labeled coverslips with 2% PFA in PBS for 5 min at 37°C. Finally, the coverslips were washed five times in washing buffer and stored at 4°C in PBS until imaging (no more than 10 days).

**SMLM imaging equipment and data collection.** Imaging was performed on a Leica DMi8 inverted microscope mounted on an antivibration table (TMC, USA), using a Leica HC PL APO 100× 1.47 numeric aperture (NA) oil immersion total internal reflection fluorescence (TIRF) objective and fiber-coupled laser launch (405 nm, 488 nm, 532 nm, 561 nm, and 642 nm) (Roper Scientific, Evry, France). The fluorescence signal was collected with a sensitive Evolve electron-multiplying charge-coupled-device (EMCCD) camera (Teledyne Photometrics). The coverslips bearing the fixed bacterial cells were mounted on a Ludin chamber (Life Imaging Services), and 600 $\mu$L of imaging buffer was added. For PALM imaging, the buffer was PBS. For dSTORM imaging, the buffer contained both an oxygen scavenger (glucose oxidase) and a reducing agent (2-mercaptoethylamine), and another 18-mm coverslip was added on top of the chamber to minimize oxygen exchanges during the acquisition.

Image acquisition and control of the microscope were driven through Metamorph (Molecular devices) in streaming mode using a 512- by 512-pixel region of interest with a pixel size of 160 nm. Image stacks typically contained 6,000 to 20,000 frames, acquired at a frequency of 33 Hz for PALM and 50 Hz for dSTORM. The power of the 405-nm laser was adjusted to control the density of single molecules per frame, keeping the 642-nm laser intensity constant. For dual-color imaging, dSTORM was performed first, followed by PALM. To limit manipulation and the potential resulting drift, coverslips were kept in dSTORM imaging medium during the PALM acquisition.

**SMLM data processing and analysis.** The PALMTracer plugin for MetaMorph (94–96) was used to process image stacks with a specific intensity threshold for each data set, to enable the generation of the localization tables. From these tables, superresolved images were generated with a pixel dimension of 40 nm. The pointing accuracies of the PALM imaging experiments were determined by tracking an individual fluorophore's motion using PALMTracer. For each track with a speed below 0.01 $\mu$m$^2 \cdot$ s$^{-1}$, the value for half of the root of the MSD was calculated (66). PALM experiments are done with a pointing accuracy of 41 nm on average. The resolution of dSTORM imaging experiments was determined by Gaussian fitting of the signals of individual fluorophores attached to the coverslip. Both resolution (66 nm) and pointing accuracy (25 nm) were extracted.

Clustering of the localizations was performed by Voronoï segmentation using SR-Tesseler (97). Object detection and clustering characterization were done using a density factor of 1 and a minimum cluster area of 0.05 pixel$^2$. The cell counter plugin of Fiji was used to count the number of cells present in our acquisition field. Dot plots were graphed using PlotsOfData (98).

## SUPPLEMENTAL MATERIAL

Supplemental material is available online only.
**SUPPLEMENTAL FILE 1**, PDF file, 1.3 MB.

## ACKNOWLEDGMENTS

We thank J. B. Sibarita, C. Butler, A. Kechkar and F. Levet for providing access to the PALMTracer plugin and SR-Tesseler software, M. Mondin for her assistance during our first SMLM tests at the Bordeaux Imaging Center, T. Ipoutcha for his assistance with *M. gallisepticum* transformation, and N. Bourg and P. Barthelemy at Abbelight for fruitful discussion. We acknowledge C. Rouveyrol for her initial work on mycoplasma sample preparation. Y.A. is personally indebted to C. Hunsa-Kredeinde for her kind review of an early draft of the manuscript.

This study was funded by the French National Agency for Research (ANR) through grants number ANR-17-CE35-0002-01 DACSyMy and ANR-21-CE44-0002 ENIgMA.

Funding acquisition, E.H. and Y.A.; project administration, Y.A.; supervision, E.H. and Y.A.; conceptualization, F.R., E.H., and Y.A.; resources, E.V. and E.H.; investigation, F.R., A.V., P.B., E.H., and Y.A.; formal analysis, F.R. and E.H.; visualization, F.R., E.H., and Y.A.; writing, F.R., E.H., and Y.A.

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
