## [Reviewer comments · Microbiology Spectrum]

Microbiology Spectrum

Imaging minimal bacteria at the nanoscale: a reliable and versatile process to perform Single Molecule Localization Microscopy in mycoplasmas

Fabien Rideau, Audrey Villa, Pauline Belzanne, Emeline Verdier, Eric Hosy, and Yonathan Arfi

Corresponding Author(s): Yonathan Arfi, University of Bordeaux

Review Timeline:

Submission Date:	February 22, 2022
Editorial Decision:	April 6, 2022
Revision Received:	April 21, 2022
Accepted:	May 10, 2022

Editor: Cezar Khursigara

Reviewer(s): Disclosure of reviewer identity is with reference to reviewer comments included in decision letter(s). The following individuals involved in review of your submission have agreed to reveal their identity: Makoto Miyata (Reviewer #1)

Transaction Report:

DOI: <https://doi.org/10.1128/spectrum.00645-22>

April 6, 2022

Dr. Yonathan Arfi
University of Bordeaux
UMR 1332 BFP
71 avenue Edouard Bourloux
CS20032
Villenave d'Ornon 33140
France

Re: Spectrum00645-22 (Imaging minimal bacteria at the nanoscale: a reliable and versatile process to perform Single Molecule Localization Microscopy in mycoplasmas)

Dear Dr. Yonathan Arfi:

Two experts have reviewed your manuscript and although they both agree that it has significant merit, it requires modifications before it can be published in Microbiology Spectrum. Please carefully consider and respond to all comments when submitting your revised manuscript.

Link Not Available

Sincerely,

Cezar Khursigara

Journals Department
Reviewer comments:

Reviewer #1 (Comments for the Author):

This study applied supramolecular microscopy to the authors' theme, MIBMIP system in class Mollicutes bacteria. The goals are clear, the advanced technology is used appropriately, and the paper is clearly written. The conclusions obtained are not very exciting, but clearly worth reporting. The authors should explain more about the biological aspects and the advanced technology

used. Details are given below.

L28: Here, we describe an efficient and reliable protocol to perform Single-Molecule Localization Microscopy (SMLM) imaging in mycoplasmas....

L114: Techniques such as Photo-Activated Localization Microscopy (PALM), Stochastic Optical Reconstruction Microscopy (STORM) or Points Accumulation for Imaging in Nanoscale Topography (PAINT) can typically yield images with 117 lateral resolutions of ~10-50 nm.

Q) The differences between PALM and dSTORM are not clear. Explain this point.

L109:and electron microscopy resolution (~1 nm).

Q) The potential resolution of electron microscopy is not 1 nm. The authors need to add some constraints.

L130: ... the cytoplasmic domain of an atypical F-type ATPase complex,....

L328: We were particularly interested in localizing an atypical F type ATPase called "F1-like-X0" (76)

Q) Explain a little more. Otherwise, authors cannot have any images on the author's goal.

L174: Mmc 0582-HA in which an HA-tag coding sequence is fused to the C-terminus of the coding sequence of MMCAP2_0582; and Mmc mEos3.2-0575 in which the codon-optimized mEos3.2 coding sequence is fused to the N-terminus of the coding sequence of MMCAP2_0575.

Q) Give sequence information to repeat experiments.

L209: Validated coverslips were stored at 4{degree sign}C in PBS until...

Q) Show the PBS contents, because many versions exist for PBS.

L243: Pointing accuracies of the PALM imaging...

L345: In order to estimate the accuracy of our detections, we used PALM-tracer to track individual emitters and calculate the Mean squared displacement (MSD) of the trajectories (Figure 2D, inset).

Q) Explain more about pointing accuracies.

L270: A good sample is.....

Q) What are the characters of a good sample?

L329: Leveraging the genome engineering tools available for this species, we generated a mutant strain Mmc mEos3.2-0575 in which the fluorescent protein is fused to the N-terminus of the β -subunit of the F1-like domain.

Q) This study does not show the influence of protein fusion on the structure and functions of the ATPase. Discuss this concern at somewhere.

L291: The resulting plasmid pMT85-PSynMyco-mEos3.2 (Figure 1A) was transformed in six....

Q) Cells were transformed, not plasmid.

L303:super-resolved images are reconstructed and analyzed by automatic Voronoï-based segmentation of these localizations (Figure 1C, Figure S2).

Q) "Voronoi tessellation" is not popular even in SMLM. The current explanation is not enough. Show purpose, principle, advantage, previous success etc.

L372: This is in accordance with the known localization of the protease at the cell surface.

Q) Explain "the known localization".

L397: This experiments clearly demonstrate the ability of super resolution technique to discriminate two types of organization in the minute mycoplasma cells, while classical microscopy failed.

Q) Do authors say that the protease and the ATPase do not colocalize? If so direct statements are acceptable. Also add these statements in abstract and other places.

Figure 1C left

Q) If there images are phase contrast, the image quality is not good compared to previous related papers. The images here look off focus. Explain the reason.

L508: References

Fix capitalization of title words and italicization of scientific names.

Minor modifications for English

L28: before

L107: has seen

L122: are have been

L171: Accurate edition
L222: set up
L224: TIRF (needs full spelling)
L226: EMCCD (needs full spelling)
L364: Coverslips samples

Reviewer #2 (Comments for the Author):

I really liked this paper. The techniques, data and explanations are all clear to me and should be to any reader. I urge swift publication. As a working mycoplasmatologist I am very pleased at both the demonstration of the SMLM techniques on mycoplasmas from all mycoplasma taxonomic groups as well as the imaging of the Mmc proteins.

I list only a tiny number of very minor suggestions that can be dealt with very quickly.

Page 2 line 53 In the sentence "These organisms derive from a common ancestor with Firmicutes through a degenerative evolution that has led to an extreme reduction in genome size (~0.6-1.35 Mbp)." The phrase "with Firmicutes" probably should say "within the Firmicutes taxon" or something similar.

Page 2 line 59 Mycoplasmas, i.e. members of the Mollicutes class include other cell wall-less organisms that infect plants and some m. of these are important plant pathogens. The genus name of these species however is not Mycoplasma.

Page 3 line 68 Add Breuer et al (PMID 30657448) to list of computational models of mycoplasmas.

Page 3 Line 84 The 2022 Thornburg paper in Cell reports a fully defined media for *M. mycoides* and defined media have been made for *M. pneumoniae*.

Page 3 Line 86 Reference Gibson (PMID) and Hutchison (PMID)

Page 4 Line 94 "Higher resolution techniques based on immunogold ... are not compatible with live-cell imaging." Perhaps a sentence or two in the Introduction or Discussion on the main challenges for applying super-resolution techniques to live mycoplasmas?

Page 4 Line 101 Add comma as shown "*M. genitalium*, which"

Page 4 Line 118 Change to "allowing resolution of the subcellular..."

Page 6 Line 174 A reference for HA-tag is needed and some explanation of why a C-terminal label was used for the lipoprotein and an N-terminal label for the ATPase beta subunit.

Page 9 Line 262. "However, this approach failed as cells remained planktonic and did not adhere to the glass. We thus developed an efficient and reliable centrifugation-based process to force the cells to sediment and attach to the coverslip (Figure S1)." Attachment of cells to the cover slip could depend more on washing than centrifugation? I suppose the media could block the PLL?

Page 13 Line 370 "localized in clusters that are predominantly at the periphery of the cell (Figure 3A). This is in accordance with the known localization of the protease at the cell surface." Perhaps include citation for known localization of the protease at the cell surface? How was it known, immunogold and EM? Was localization of the protease in clusters also known? Regarding clustering in super-resolution, what might serve as a negative control for clustering? In other systems, are there probes observed to distribute more uniformly?

Page 15 Line 430 "This bacterium, in addition to being the etiologic agent of Contagious Caprine PleuroPneumonia,.."
Subspecies capri and more importantly, strain GM12 are not the agents of CCPP. That is the capripneumonia subspecies, which is considered a select agent by the United States. The capri subspecies does not cause CCPP..

Page 16 Line 458 Rather than say "references", perhaps "catalog numbers" would be more appropriate.

Page 16 Line 474. "genetic edition ..." I think typo, should be "... editing ..."

Page 25 Fig 1. The purpose of the Tesseler segmentation was unclear to me in the main figures but clarified in Fig S7. This needs to be made clear without the need to go to the supplementary data.

Staff Comments:

Preparing Revision Guidelines

Please return the manuscript within 60 days; if you cannot complete the modification within this time period, please contact me. If you do not wish to modify the manuscript and prefer to submit it to another journal, please notify me of your decision immediately so that the manuscript may be formally withdrawn from consideration by Microbiology Spectrum.

Dear Dr. Cezar Khursigara, dear Reviewers #1 and #2, dear Microbiology Spectrum Editorial Staff,

We would like to thank you for the rapid reviewing of our manuscript, and for your comments and questions.

You will find below a complete, point-by-point response to your queries. Please note that for clarity your comments will be in black text, and our responses in blue text.

Reviewer comments:

Reviewer #1 (Comments for the Author):

This study applied supramolecular microscopy to the authors' theme, MIBMIP system in class Mollicutes bacteria. The goals are clear, the advanced technology is used appropriately, and the paper is clearly written. The conclusions obtained are not very exciting, but clearly worth reporting. The authors should explain more about the biological aspects and the advanced technology used. Details are given below.

We thank Reviewer #1 for the positive appraisal of our manuscript. We do agree with the notion that this article is mostly technical and does not in itself provide an “exciting” biological model. But we do believe that by paving the way towards SMLM for mycoplasmologists, our publication will help others and yield a better understanding of Mollicutes and their atypical biology. Following your detailed comments, we have modified the manuscript in order to better describe and explain technological and biological aspects that were not clear enough in our first submission.

L28: Here, we describe an efficient and reliable protocol to perform Single-Molecule Localization Microscopy (SMLM) imaging in mycoplasmas....

L114: Techniques such as Photo-Activated Localization Microscopy (PALM), Stochastic Optical Reconstruction Microscopy (STORM) or Points Accumulation for Imaging in Nanoscale Topography (PAINT) can typically yield images with

117 lateral resolutions of ~10-50 nm.

Q) The differences between PALM and dSTORM are not clear. Explain this point.

We agree with Reviewer #1 that our introduction expects from the reader to have an initial knowledge of single molecule localization microscopy techniques. As this might not be the case, we have expanded the section by a few lines in order to better describe the fundamental principles of PALM and dSTORM. We also believe that the references in this section cited are very well written introductory materials that explain thoroughly these concepts. We also note that the concepts of PALM and dSTORM principles are described at the beginning of their respective Results paragraphs (lines 275-278, and lines 357-360, respectively).

L109:and electron microscopy resolution (~1 nm).

Q) The potential resolution of electron microscopy is not 1 nm. The authors need to add some constraints.

As we have no specific background in electron microscopy, we cited the broad value of “~1 nm” as we found it in different publications. We acknowledge that this value is “generic” and does not consider the type of electron microscopy experiment performed, nor the type of sample processing. In order to solve the issue with the sentence, we have removed both indications of “resolution” (ie. “~200 nm” and “~1 nm”). The sentence now reads “[...] the rapid development of multiple new fluorescence microscopy techniques aimed at bypassing the diffraction limit and bridging the gap between optical imaging resolution and electron microscopy resolution.

L130: ... the cytoplasmic domain of an atypical F-type ATPase complex,...

L328: We were particularly interested in localizing an atypical F type ATPase called "F1-like-X0" (76)

Q) Explain a little more. Otherwise, authors cannot have any images on the author's goal.

We understand the Reviewer’s comment, as we selected a model protein that had a specific interest to our research group, but is probably obscure to the general readership. We have added details on this ATPase, its expected structure and function, at the beginning of the Results section 3.

L174: Mmc 0582-HA in which an HA-tag coding sequence is fused to the C-terminus of the coding sequence of MMCAP2_0582; and Mmc mEos3.2-0575 in which the codon-optimized mEos3.2 coding sequence is fused to the N-terminus of the coding sequence of MMCAP2_0575.

Q) Give sequence information to repeat experiments.

Coding sequences of the MMCAP2-0582-HA and MMCAP2-0575-mEos3.2 fusions have been added in Supplementary Materials.

L209: Validated coverslips were stored at 4{degree sign}C in PBS until...

Q) Show the PBS contents, because many versions exist for PBS.

We thank Reviewer 1 for pointing out this missing information. PBS was NaCl 137 mM - KCl 2.7 mM - Na₂HPO₄ 10 mM - KH₂PO₄ 1.8 mM. This is now mentioned in the text.

L243: Pointing accuracies of the PALM imaging...

L345: In order to estimate the accuracy of our detections, we used PALM-tracer to track individual emitters and calculate the Mean squared displacement (MSD) of the trajectories (Figure 2D, inset).

Q) Explain more about pointing accuracies.

The pointing accuracy is our ability to localize precisely a protein. The pointing accuracy is dependent of the technique (STORM is better than with PALM) and the setup. There is various way to calculate the pointing accuracy. Here, we decided for PALM experiments to determine the detection distribution of each individual fluorophore. The lifetime of emission of the mEOS3.2 is around 500 ms, as acquisition are at video rate, each emitter is followed for tens of images. The localization of the fluorophore on each image is extracted by wavelet analysis then a trajectory is obtained by reconnecting all detection from the same emitter in function of time. As we are working on fixed

tissue, the variation of the coordinates is only based on the pointing uncertainty. The specific analysis of the distribution gave us access to the pointing accuracy (which is different of the resolution which calculate the closest distance where two emitters can be separated).

L270: A good sample is.....

Q) What are the characters of a good sample?

We are not sure to understand the Reviewer's comment, as the sentence in question reads "A good sample is characterized by bacterial cells deposited in a monolayer, regularly spaced and separated from each other by a few microns". The notion of "good sample" indicates that the prepared coverslip: i) indeed contains cells, ii) that the cells form a mono-layer, iii) that the cells are sufficiently separated from one another. These characteristics will then allow for a successful image collection.

L329: Leveraging the genome engineering tools available for this species, we generated a mutant strain Mmc mEos3.2-0575 in which the fluorescent protein is fused to the N-terminus of the β -subunit of the F1-like domain.

Q) This study does not show the influence of protein fusion on the structure and functions of the ATPase. Discuss this concern at somewhere.

We agree with the Reviewer's comment that we do not make any comment on the functionality of our β sub-unit fusion. This question is systematic when dealing with fusion proteins, as the impact of the fusion on expression, folding and stability is hard to predict. We believe that this concept is now well-known in the microbiologist community.

In our case, we do not have a test that can assess whether the ATPase is functional in itself. However, during earlier studies on this ATPase, we have shown that we can add a small Strep-tag II tag at the N-terminus of MMCAP2_0575 (β sub-unit). We can then use this tag to perform an affinity purification of the F1-like complex. We then used single particle cryoEM to purify the complex and showed that it is correctly assembled in the cells and exhibits its expected $3\alpha3\beta-1\epsilon-1\gamma$ architecture. Thus, when we had to select a candidate for mEos3.2 fusion, we elected to test the Nter of MMCAP2-0575 as it appeared amendable to fusion without perturbation of the F1-like domain. We understand that mEos3.2 is a much bigger fusion than Strep-tag II, and thus could have a strong negative impact on the ATPase assembly/function. This is mitigated by several factors:

- We were able to generate a Nter fusion of mEos3.2 to MMCAP2_0575, and the Nter fusion protein was detected by western blot and appeared stable in the cell over time.
- The mEos3.2 is fused to MMCAP2_0575 through a glycine-serine linker, in order to give some degrees of freedom and flexibility to the fluorescent protein, and limit potential perturbations to the beta sub-unit
- Although we have not found examples of fusions between the F1F0 ATPase F1 domain and a fluorescent protein, several studies have grafted a biotin binding tag to the beta subunit N-terminus, followed by biotinylation and interaction with streptavidin-coated gold nanoparticles (PMID: 23535652) or actin filaments (PMID: 12357031). These studies demonstrate the ability of the beta sub-unit N-terminus to accommodate very large molecular assemblies without perturbation to the ATPase functionality.

Overall, and based on the elements cited above, we are confident that our fusion protein is functional. We have added a section of discussion in the paragraph about PALM indicating to the

reader that fluorescent protein fusion carry with them the potential to unwillingly damage the protein of interest and to compromise subsequent analyses.

L291: The resulting plasmid pMT85-PSynMyco-mEos3.2 (Figure 1A) was transformed in six....

Q) Cells were transformed, not plasmid.

Sentence was corrected to "The resulting plasmid pMT85-PSynMyco-mEos3.2 (Figure 1A) was used to transform six Mycoplasma species"

L303:super-resolved images are reconstructed and analyzed by automatic Voronoï-based segmentation of these localizations (Figure 1C, Figure S2).

Q) "Voronoi tessellation" is not popular even in SMLM. The current explanation is not enough. Show purpose, principle, advantage, previous success etc.

The application of Voronoï tessellation to segment/cluster SMLM data is a relatively recent development, with publications describing the concept and associated software starting in 2015 (PMID: 26344046, PMID: 27068792, PMID: 31147535, PMID: 29635310). Multiple alternative SMLM data analysis strategies have been described to estimate local densities, for segmentation and cluster analysis methods such as Ripley's L function, median or Gaussian filtering of histogram images, k-means and DBSCAN. Our choice of SR-Tesseler, a Voronoï tessellation-based software, was guided predominantly by three factors:

- 1) The manuscript presenting SR-Tesseler has been published in a high-quality peer-reviewed journal, and has been highly cited in a short amount of time.
- 2) The developers of SR-Tesseler are part of the same research group as one of the authors, giving us quick access to their expertise.
- 3) The ability of the software to run efficiently on regular, low power computers while still providing precise, robust and automatic quantification of protein organization at different scales, from the cellular level down to clusters of a few fluorescent markers.

We do not infer that this approach is optimal, nor devoid of drawbacks. It is an analysis solution amongst other that worked well in our analysis pipeline. In order to better inform the reader on the availability of other strategies, we have added a new discussion section on data analysis, in which we reference multiple reviews on the matter.

L372: This is in accordance with the known localization of the protease at the cell surface.

Q) Explain "the known localization".

We have previously studied the protease MIP, and demonstrated that this protein is found at the surface of the mycoplasma cells where it acts as an antibody-degrading system. These findings are reported in a recent publication (PMID: 33674316). In order to be more explicit, the sentence has been modified and now reads "This is in accordance with the previously demonstrated localization of the protease at the cell surface ([reference to PMID: 33674316])."

L397: This experiments clearly demonstrate the ability of super resolution technique to discriminate two types of organization in the minute mycoplasma cells, while classical microscopy failed.

Q) Do authors say that the protease and the ATPase do not colocalize? If so direct statements are acceptable. Also add these statements in abstract and other places.

No, this is not what we inferred by this sentence. We wanted to convey the idea that we had successfully performed two color imaging, and that each color channel gave us access to organization data that were not accessible by classical microscopy: the cytoplasmic mEos filling the cytoplasm, and the MIP protease being found in a small number of distinct clusters scattered across the cell surface. We note that the question by Reviewer #1 suggest a misunderstanding: our two-colors experiments were not performed with the ATPase and the protease, but with the protease and the mEos free in the cytoplasm (not in fusion). This is indicated a few lines above in the sentence "To do so, the strain Mmc 0582-HA was transformed with the plasmid pMT85-PSynMyco-mEos3.2." This is due to the fact that obtaining double mutants in mycoplasmas is tedious, and thus we do not yet have a MMCAP_0575-mEos-Nt + MMCAP2_0582-HA-Ct mutant. Thus, we elected to transform our pMT85-mEos plasmid in a cell bearing the mutation MMCAP2-0582-HA-Ct to perform the two-color imaging. Transformation is fast in Mmc, and we had all the necessary material on hand.

Figure 1C left

Q) If there images are phase contrast, the image quality is not good compared to previous related papers. The images here look off focus. Explain the reason.

These images are indeed phase contrast, and they are indeed "not very good" and "blurry". Sadly, this is the best we could do on our SMLM set up. The poor quality is due to multiple factors:

- 1) Phase contrast images had to be zoomed-in to match the scale of the super-resolved images. As our camera has a relatively low resolution (512x512 px), zooming will produce a grainy, pixelated aspect. Mycoplasma cells are also extremely small (sub-micron in their largest dimension), so we had to zoom a lot.
- 2) Image acquisition was performed with a 100X objective. Higher magnification objectives were available (160X), but with lower numerical aperture. We elected to have a higher NA to collect more photons and have the potential for higher resolution in SMLM. The phase contrast is only used to show the reader that we are indeed "looking at a cell".
- 3) Mycoplasma do not have a cell wall and tend to generate very low contrast images in "contrast phase".

L508: References

Fix capitalization of title words and italicization of scientific names.

References have been parsed, and we hope that we have fixed all mistakes. We also hope that any remaining issue will be dealt with during proof reading, should our revised manuscript satisfy the Reviewers and the Editor.

Minor modifications for English

L28: before

L107: has seen

L122: are have been

L171: Accurate edition

L222: set up

L224: TIRF (needs full spelling)

L226: EMCCD (needs full spelling)

L364: Coverslips samples

Text has been edited accordingly

Reviewer #2 (Comments for the Author):

I really liked this paper. The techniques, data and explanations are all clear to me and should be to any reader. I urge swift publication. As a working mycoplasmaologist I am very pleased at both the demonstration of the SMLM techniques on mycoplasmas from all mycoplasma taxonomic groups as well as the imaging of the Mmc proteins.

We would like to thank Reviewer #2 for these very kind words and positive appraisal of our work. We hope that most mycoplasmaologists will benefit from our work, and that it will facilitate the adoption of cutting-edge microscopy methods to the field.

I list only a tiny number of very minor suggestions that can be dealt with very quickly.

Page 2 line 53 In the sentence "These organisms derive from a common ancestor with Firmicutes through a degenerative evolution that has led to an extreme reduction in genome size (~0.6-1.35 Mbp)." The phrase "with Firmicutes" probably should say "within the Firmicutes taxon" or something similar.

We have modified the sentence following the suggestion of Reviewer #2.

Page 2 line 59 Mycoplasmas, i.e. members of the Mollicutes class include other cell wall-less organisms that infect plants and some m. of these are important plant pathogens. The genus name of these species however is not Mycoplasma.

We agree with Reviewer #2 that there are some semantic disagreements on what a "mycoplasma" is. This is the issue with colloquial names. Some phytoplasmaologist colleagues would really dislike being told that they work on "mycoplasmas". In addition, as phytoplasmas are currently uncultured in laboratory conditions, and not amenable to genetic engineering, they are probably outside the scope of this manuscript (as we only showcase results on members of the *Mycoplasma* genus).

Page 3 line 68 Add Breuer et al (PMID 30657448) to list of computational models of mycoplasmas.

Suggested reference has been added.

Page 3 Line 84 The 2022 Thornburg paper in Cell reports a fully defined media for *M. mycoides* and defined media have been made for *M. pneumoniae*.

We agree with the Reviewer #2 comment regarding defined media for *M. pneumoniae*, but were not aware of the indicated publication for *M. mycoides*. We have slightly edited the sentence to be broader. It now reads “these organisms are slow growing and require complex and often undefined culture media”.

Page 3 Line 86 Reference Gibson (PMID) and Hutchison (PMID)

We are not sure of what the Reviewer #2 is requesting here. References to Gibson and Hutchison are made in the text two paragraphs above in the section on synthetic biology (sentence “Mycoplasmas have been at the center of landmark studies such as the production of the first cell governed by a chemically synthesized genome, and later of the first minimal synthetic bacterial cell (12, 13).”). Reference 12 & 13 correspond to Gibson *et al* and Hutchison *et al* papers.

In the sentence line 86, we refer to the genome editing tools from Tsarnopoulos *et al*, which derive from Gibson and Hutchison work.

Page 4 Line 94 "Higher resolution techniques based on immunogold ... are not compatible with live-cell imaging." Perhaps a sentence or two in the Introduction or Discussion on the main challenges for applying super-resolution techniques to live mycoplasmas?

An earlier version of the manuscript had a Discussion section dedicated to live imaging. This section was removed during redaction, mostly to fit within the word-limit request by the journal. We have added back this section in the Discussion, which reads “*It is noteworthy that our PALM experiments were performed on fixed samples, and thus did not leverage an important benefit of this technique: the ability to image live cells (84). This choice was constrained by biosafety requirements, as most mycoplasmas are BioSafety Level 2 organisms. Live-cell imaging would have required the microscopy set-up to be located in an appropriate laboratory, which was not available to us. This issue will probably affect most mycoplasmaologists as BSL2-confined SMLM set-up are currently rare worldwide. In addition, SMLM microscopes are predominantly housed in laboratory working with eukaryotic cell cultures. As mycoplasmas are frequent contaminants of these cultures (85), mycoplasmaologist might be barred from accessing these facilities with live cells. Transition to live-cell PALM is nonetheless desirable, as it would enables imaging of dynamics processes, offsetting the relative lack of resolution of the technique by providing more physiologically-relevant data. Live cell imaging will also require an alternative sample preparation process, as adhering to the poly-L-lysine coated coverslip might have a negative impact on the cell growth of normally planktonic species. Protocols revolving around sandwiching bacterial cells between agarose pads are well documented (83 , 86) and could be rapidly adapted to mycoplasmas.*”

This section is “plug and play” and can be removed if deemed to long or not appropriate by the Editor or the Reviewers.

Page 4 Line 101 Add comma as shown "M. genitalium, which"

Text has been edited as suggested.

Page 4 Line 118 Change to "allowing resolution of the subcellular..."

Text has been edited as suggested.

Page 6 Line 174 A reference for HA-tag is needed and some explanation of why a C-terminal label was used for the lipoprotein and an N-terminal label for the ATPase beta subunit.

A reference (PMID: 2455217) has been added for the HA-tag, as well as a sentence explaining the rationale for the tag localization in Ct, and the Nt fusion of mEos.

Page 9 Line 262. "However, this approach failed as cells remained planktonic and did not adhere to the glass. We thus developed an efficient and reliable centrifugation-based process to force the cells to sediment and attach to the coverslip (Figure S1)." Attachment of cells to the cover slip could depend more on washing than centrifugation? I suppose the media could block the PLL?

We agree with the Reviewer that media composition could be a critical factor in preventing binding to poly-lysine, as mycoplasma growth media contains high concentrations of proteins for serum which can interact with the poly-lysine. When we centrifuge cells suspended in media onto coverslips, we see attachment, but usually at low densities and with a poorly reliable yield. Washed cells, suspended in a saline buffer, are not susceptible to this blocking by medium content. However, during some of our tests, we tried to incubate the glass slides with washed cells in buffer and failed to see any significant attachment. Centrifugation was the only way to systematically generate well adhered cells, at good densities. We also note that we mistakenly performed the experiment on non-functionalized coverslips (no poly-lysine) and got very nice preparations out of it.

Page 13 Line 370 "localized in clusters that are predominantly at the periphery of the cell (Figure 3A). This is in accordance with the known localization of the protease at the cell surface." Perhaps include citation for known localization of the protease at the cell surface? How was it known, immunogold and EM? Was localization of the protease in clusters also known? Regarding clustering in super-resolution, what might serve as a negative control for clustering? In other systems, are there probes observed to distribute more uniformly?

Citation of the work in which the protease localization has been established has now been added (as this was also a comment by Reviewer #1). Based on both Reviewers' comment, we believe that our initial phrasing was not clear enough.

We have previously shown that the protease is bound to the cell surface by a transmembrane domain (see PMID: 33674316) and that the protease is on the outside face of the membrane (as its specific antibody cleavage activity was detected on intact cells). The dSTORM data show a localization of the signal at the periphery of the cell, which corresponds to what we expected from a protein located at the surface of the bacteria.

Meanwhile, the term “clusters” here refers to clusters in detected signal (as sentence reads “dSTORM datasets reveal that the fluorescence signal is localized in clusters that are predominantly at the periphery of the cell”). This is not an indication that the protease itself forms clusters (i.e: multiple copies of the protein adjacent to one another). Just that the signal we detect is found in a small subset of localization, and this that our detections form clusters. We cannot currently correlate “detection clusters” with “protein cluster”. As we perform a primary-secondary antibody labelling, it is probable that a single MIP protease is eventually bound to multiple fluorescent antibodies, and this yields many adjacent detections. Signal processing and data analysis can be performed to quantify the average detection numbers given by a single fluorescent antibody (often found on the coverslip, as “background noise”; and use this value to infer the number of target proteins at a given location. We have elected not to perform this analysis, as our double labeling method introduces too much uncertainty. We are currently developing tools (namely nanobodies) to localize proteins of interest with a single probe and a 1:1 probe to protein of interest ratio, and a 1:1 fluorophore to probe ratio. These developments will then allow us to map precisely where each protein is localized, and establish the existence or not of protein clusters.

Page 15 Line 430 "This bacterium, in addition to being the etiologic agent of Contagious Caprine PleuroPneumonia,..". Subspecies capri and more importantly, strain GM12 are not the agents of CCPP. That is the capripneumonia subspecies, which is considered a select agent by the United States. The capri subspecies does not cause CCPP..

I can only apologize for this mistake, and thank the Reviewer for pointing it out. CCPP is indeed not caused by *Mmc*. I have no idea why this sentence was written. The text has been edited and now reads “This bacterium, in addition to being the etiologic agent of pulmonary infections in goats, has emerged as a major model in both synthetic and fundamental biology.”

Page 16 Line 458 Rather than say "references", perhaps "catalog numbers" would be more appropriate.

The text has been edited according to the suggestion.

Page 16 Line 474. "genetic edition ..." I think typo, should be "... editing ..."

Indeed, these are improper translation from French to English. “edition” has been replaced by “editing” here and elsewhere in the manuscript.

Page 25 Fig 1. The purpose of the Tessler segmentation was unclear to me in the main figures but clarified in Fig S7. This needs to be made clear without the need to go to the supplementary data.

We understand the Reviewer complaint, and we agree that the concepts behind SR-Tessler might be a bit obscure to the readers. However, we are uncertain of how to proceed with this request. We have thought for a long time about figures compositions, and we finally settled on reserving main text figures for scientific data, and use the supplementary figures to provide schematics explaining our workflow (sample preparation, data processing, tessellation, etc...). Trying to shift some of the S.I figures to the main text would significantly extend the length of the paper and the size of the figures.

We are open to this solution (I personally would prefer that SI did not exist at all), but we believe that this decision should be made by the Editor. In the meantime, we would like to keep the figures as they are.

May 10, 2022

Dr. Yonathan Arfi
University of Bordeaux
UMR 1332 BFP
71 avenue Edouard Bourlaux
CS20032
Villenave d'Ornon 33140
France

Re: Spectrum00645-22R1 (Imaging minimal bacteria at the nanoscale: a reliable and versatile process to perform Single Molecule Localization Microscopy in mycoplasmas)

Dear Dr. Yonathan Arfi:

Your manuscript has been accepted, and I am forwarding it to the ASM Journals Department for publication. You will be notified when your proofs are ready to be viewed.

Sincerely,

Cezar Khursigara
Editor, Microbiology Spectrum
